# A Novel Perspective on Ischemic Stroke: A Review of Exosome and Noncoding RNA Studies

**DOI:** 10.3390/brainsci12081000

**Published:** 2022-07-28

**Authors:** Qianwen Wang, Yuhui Chen, Lingbing Meng, Jiawen Yin, Li Wang, Tao Gong

**Affiliations:** 1Department of Neurology, Beijing Hospital, National Center of Gerontology, Institute of Geriatric Medicine, Chinese Academy of Medical Sciences, No. 1 DaHua Road, Dong Dan, Beijing 100730, China; jintianqing117@126.com (Q.W.); cmucyh@163.com (Y.C.); yinjiahun@gmail.com (J.Y.); wangliorwl@163.com (L.W.); 2Graduate School, Chinese Academy of Medical Sciences and Peking Union Medical College, No. 9 Dongdansantiao, Dongcheng District, Beijing 100730, China; bjh_meng0505@student.pumch.edu.cn; 3Department of Cardiology, Beijing Hospital, National Center of Gerontology, Institute of Geriatric Medicine, Chinese Academy of Medical Sciences, No. 1 DaHua Road, Dong Dan, Beijing 100730, China

**Keywords:** exosomes, lncRNA, circRNA, ischemic stroke, diagnostics, therapeutics

## Abstract

Ischemic stroke is a life-threatening condition that also frequently results in long-term disability. Currently, intravenous thrombolysis with tissue plasminogen activator and mechanical thrombectomy is the most popular treatment. However, the narrow time window and related complications limit the treatment benefits. Exosomes have recently emerged as ideal therapeutic candidates for ischemic stroke with the ability to pass through the blood_brain barrier and mediate intercellular communication, in addition, exosomes and their contents can be bioengineered to implement targeted delivery. In the last two decades, exosomes and exosomal noncoding RNAs have been found to be involved in the pathophysiological progression of ischemic stroke, including atherosclerosis, apoptosis, inflammation, oxidative stress, and neurovascular remodeling. In this review, we describe the latest progress regarding the role of exosomal long noncoding RNAs and circular RNAs in the occurrence, progression, and recovery of ischemic stroke. Exploration of exosomal noncoding RNAs and their correlated effects in ischemic stroke may facilitate accurate diagnosis, and they may serve as new therapeutic targets for the disease.

## 1. Introduction

Stroke is the second most common cause of death worldwide and the third-leading cause of death and disability combined, and ischemic stroke represents almost 87% of all stroke incidents [1,2]. Thrombolysis with tissue plasminogen activator and endovascular thrombectomy is currently the most popular treatment for ischemic stroke, although other treatments are available. Alteplase, as the only reperfusion drug approved by the FDA (Food and Drug Administration of the United States), has a relatively narrow beneficial time window (no more than 4.5 h) and may cause hemorrhagic transformation due to low targeting effects on thrombi and ischemia_reperfusion injury resulting from the over-production of reactive oxygen species (ROS), limiting its clinical application. Endovascular thrombectomy needs to take place within 24 h of the onset of stroke, which is not always feasible; furthermore, reclusion and distal fragmentation are also important challenges to consider [3]. Therefore, there is certainly a clinical demand for the investigation of novel therapeutic regimens for the treatment of ischemic stroke. In the past two decades, exosomes and noncoding RNAs have both gained increased traction in the initiation and development of such treatments.

Exosomes are a type of extracellular vesicles (EVs) originating from endosomes, which can be secreted by several cell types and detected in a number of biological fluids under normal physiological and pathological conditions [4]. Exosomes are of endosomal origin with a diameter of approximately 40–160 nm (averaging approximately 100 nm) and are generated by the double invagination of the plasma membrane with subtle manipulation involving the endoplasmic reticulum, trans-Golgi network, and mitochondria. Specifically, the first invagination of the plasma membrane through endocytosis contributes to the formation of early-sorting endosomes (ESEs); then, ESEs mature into late-sorting endosomes (LSEs) and multivesicular bodies containing intraluminal vesicles (future exosomes). This process is regarded as the second invagination of the endosomal membrane [5]. Exosomes can integrate with other organelles and intracellular vesicles during their formation process, as can their contents, which include proteins, DNA, RNA, lipids, and metabolites, making exosomes quite heterogeneous and diverse. However, questions such as whether their production and constitution are influenced by age remain. Other needs, such as those for the precise isolation and standardized purification of exosomes, are expected to continue to advance with time, though they may be underdeveloped at present. Overall, their endogenous derivation from live cells gives exosomes unique features for near- and long-distance intercellular communication such as intrinsic homing ability, high bioavailability, biocompatibility, and low immunogenicity and toxicity, all of which contribute to their possible application in the effective treatment of a variety of diseases such as cancer, neurological diseases, and cardiovascular disease. The abundant molecular payloads of exosomes are available expeditiously using minimally invasive liquid biopsies, providing appreciable value as promising circulating biomarkers for diagnostic and surveillance purposes. Furthermore, exosomes can also be engineered physically and chemically for applications in targeted delivery, allowing for an attractive method of gene therapy. Nowadays, the intravenous injection of exosomes containing specific contents can successfully reach the brain to achieve functional regulation, and multiple stem-cell-derived exosomes have been found to present recovery and regenerative abilities [6,7,8]. Exosome-mediated measures that could become practical in stem cell transplantation are intriguing because they involve fewer complicating concerns compared to transplantation directly [9]. Even so, we believe that it is worthwhile to examine the diverse applications of exosome biology in the diagnosis, therapy, and monitoring of disease.

Several studies have focused on exosome-associated RNAs and have found that exosomes might be ideal carriers for the transmission of molecular information and intercellular communication due to their ability to protect their contents from RNase degradation. In the experiment of Valadi, exosomal mRNA was translated to proteins after successful transfer to recipient cells, and Pegtel’s study showed that miRNAs could be delivered through exosomes to receptors and could exert a biological function [10,11]. In more recent years, other RNA types have been identified in exosomes, including long noncoding RNA (lncRNA), circular RNA (circRNA), small nucleolar RNA (snoRNA), and small interfering RNA (siRNA). In addition to conventional extraction and detection tools for exosomal nucleic acids such as Droplet PCR, Microfluidics for On-Chip Extraction and Detection, Ion-Exchange Nanodetectors, and LSPR-Based Assays, researchers have been gradually developing new techniques, offering new clinical opportunities for further personalized molecular medicine. Related research has found that increasingly different forms of exosomal RNAs play comprehensive roles in diverse human diseases including ischemic stroke in particular [12,13].

This review considers the latest research in order to illustrate the widespread application of exosomes and exosomal noncoding RNAs in the diagnosis, surveillance, and therapeutic efficacy of ischemic stroke, ranging from the differential expression of exosomal RNAs under disease conditions, to the diverse biological effects of exosomal RNAs in the progression of ischemic stroke and to the therapeutic potential of exosomes and their contents in the treatment of ischemic stroke. We describe recent advances in exosomal RNAs mainly lncRNAs and circRNAs in the diagnosis and treatment of ischemic stroke, and present the possibility of using novel exosome-based gene therapy in clinical application.

## 2. Methods and Inclusion Criteria

We utilized the electronic databases PubMed, Web of Science, and EMBASE to identify suitable studies published prior to April 2022 using specific keywords following the search strategies: (1) Ischemic stroke OR Stroke OR Cerebrovascular disease, (2) Exosome OR Exosomal OR Extracellular vesicles, (3) RNA. Some articles were identified via manual screening of relevant references from other studies on the subject.

Two authors assessed retrieved records independently for eligibility, and disagreements were resolved by consensus. The included research mainly contained the basic exploration of exosomes or exosomal noncoding RNAs and ischemic stroke. We excluded studies of other cerebrovascular diseases such as hemorrhagic stroke and Moya Moya disease.

## 3. The Potential Diagnostic Value of Exosomal Noncoding RNAs in Ischemic Stroke

An array of evidence has revealed that exosomes may be ideal biomarkers for different diseases, including ischemic stroke, due to their convenient acquisition and widespread distribution in diverse biological fluids. Previous studies have detected that various noncoding RNAs are differentially expressed under hypoxic and hypovolemic states in animal models and patients with ischemic stroke and they could be used as monitoring factors in the progression of ischemic stroke [14,15]. In this section, we describe the altered expression profile of exosomal noncoding RNAs in the disparate models of ischemic stroke (Table 1).

### 3.1. The Expression Profile of Exosomal Long Noncoding RNAs

The team of Xu reported the differential expression of exosomal lncRNAs in acute minor stroke patients whose NIHSS (National Institute of Health stroke scale) scores were no more than 4 compared with healthy controls. One hundred patients with symptoms appearing within 24 h, paired with 100 controls, were enrolled in the study and blood samples were obtained without any treatment. In their RNA sequencing (RNA-seq) analysis, containing three-pair subjects, 1096 exosomal lncRNAs had significantly altered expression levels (>1.5-fold, *p* < 0.05), including 307 with increased expression and 789 with decreased expression. After this, they selected the neurotrophin signaling pathway and four relevant lncRNAs (upregulated: lnc-CRKL-2, lnc-NTRK3-4; downregulated: RPS6KA2-AS1, lnc-CALM1-7) according to GO (Gene Ontology) enrichment analysis and KEGG (Kyoto Encyclopedia of Genes and Genomes) pathway analysis for next-step validation. The results of RT-qPCR (quantitative real-time polymerase chain reaction) testing of these four lncRNAs showed consistent trends with their preceding experiment between exosomes from patients and controls [16]. Their research illustrated that exosomal lncRNAs may be a viable option for the early detection of the disease, but further exploration of the molecular bases and underlying signal pathways is still needed.

Soon after these results were presented, Zhang et al. [17] performed a comprehensive exosomal lncRNA sequencing with related networks in order to explore their diagnostic potential and underlying mechanisms in large-artery atherosclerotic (LAA) stroke. Thirty-five LAA patients (>50% extracranial or intracranial arteries stenosis) within 72 h of stroke onset and an equal number of controls were enrolled in their study, and all subjects were matched by clinical characteristics, including age, sex, and vascular risk factors. Initially, they found that 1060 lncRNAs were differentially expressed, covering 226 lncRNAs that were upregulated and 794 that were downregulated significantly in the LAA cases versus controls using the high-throughput sequencing (HTS) method (5 paired samples). The chromosomal distribution of differentially expressed genes showed differentiated transcripts spread over every chromosome and chromosome 1 contained the most. Based on the target gene functions and differential expression, 8 hub lncRNAs relevant to atherosclerosis were selected in order to construct related networks according to their reads and changed folds, and biological function analysis of these networks by GO and KEGG pathways showed that inflammation, immunity, cell adhesion, and apoptosis may be attributed to the progression of atherosclerosis. After this analysis, the authors validated the observed differential expression from the HTS results of 21 hub genes involved in the lncRNA-related networks by a quantitative real-time PCR (qRT-PCR) experiment in a cohort of stroke patients (30 stroke versus 30 control). In the validation set, they constructed an exo-lncRNA-related diagnostic model of LAA stroke and focused on two lncRNAs, lnc_001350 and lnc_002015, as propelling and defensive factors for the occurrence of LAA stroke (*p* < 0.05). Furthermore, the diagnostic efficacy of exo-lncRNA-related networks could be higher than any single gene in the network as the area under the curve (AUC) of the lnc_002015-related network and lnc_001350-related network were 0.959 and 0.970, respectively. They also presented results on the lnc_002015/hsa-miR-342/PTPRC/AAMP and lnc001350/hsa-miR-3127/ST14 pathways since PTPRC and miR-342 have previously been reported to mediate the progression of AS and the inflammatory response. Of note is that the two lncRNAs in question are more probably located in the cytoplasm, which would enhance the reliability of exo-lncRNA-related functional networks.

Recently, Zhang’s team affirmed the diagnostic and prognostic value of exosomal lncRNAs for LAA stroke in the clinical setting; furthermore, their research indicated that exosomal lncRNAs rather than plasmatic lncRNAs might be particular biomarkers for LAA stroke, with an obvious advantage in the AUC. They randomly assigned 602 participants to a discovery set (n = 12), validation set (n = 80), and replication set (n = 510), and RNA sequencing and qRT-PCR were used to detect the differential expression of lncRNAs. Firstly, 222 downregulated and 97 upregulated lncRNAs were discerned in the discovery set (|log_2_fold change| ≥ 2, *p* < 0.05), and functional enrichment analysis showed that the target genes of these exo-lncRNAs were mainly involved in the AS-related pathological process. After validation testing and subgroup analysis, the results indicated that upregulated exo-lnc_000048, exo-lnc_001350, and exo-lnc_016442 were consistent with an increased risk of LAA stroke. The AUC of the combined three exo-lncRNAs was 0.936, which was not influenced by the traditional biomarkers of IS (triglycerides, total cholesterol, low-density lipoprotein). Moreover, the adjusted logistic analysis revealed that the combination of exo-lnc_001350, exo-lnc_016442, and NIHSS provided better prognostic capacity for one-month unfavorable outcomes, independent of sex, age, and the above traditional biomarkers [18]. Overall, these results provided exosomal lncRNA models with satisfactory diagnostic and prognostic valuew in LAA stroke. Other multicenter and large-scale studies are warranted to verify these results, which could be meaningful for the rapid detection and preferable functional outcomes of the disease.

According to their research, we recognized the diagnostic value of plasma exosomal lncRNAs and related networks in acute minor stroke as well as LAA stroke; moreover, the diagnostic possibilities of exosomal RNAs in the different types of ischemic stroke are worth consideration.

### 3.2. The Expression Profile of Exosomal Circular RNAs

Xu et al. discovered altered plasma circRNA expression profiles in IS patients within 72 h of onset compared to healthy controls. They found a total of 3540 differentially expressed circRNAs in exosomes, including 1177 with increased expressions and 2363 with decreased expressions (fold change > 2, *p* < 0.05). Among the entirety of the abnormally expressed circRNAs, most of them were located at chromosome 2 and were less than 500 nucleotides in length; in addition, exons and introns made up over 60% of the source regions. Bioinformatics analysis revealed and predicted underlying pathways in the IS pathology, such as the MAPK signaling pathway, lipids, and atherosclerosis, neurotrophin signaling pathway, etc. Moreover, ten hub genes were selected from the protein_protein interaction network, providing clues for the subsequent mechanistic research of noncoding RNAs in ischemic stroke [19]. It should be pointed out that the NIHSS scores of patients in this study were 11–25, while mild stroke was not included; more research covering different subtypes and orders of severity for stroke is needed in the future.

In a study by Xiao et al. [20], a comprehensive examination of exosomal circRNAs isolated from the peripheral blood of LAA stroke patients was performed. There were 37 patients within 72 h of symptom onset paired with 37 normal controls in the study, and stenosis of 50% or above in intracranial or extracranial arteries was discerned in all cases. At first, the authors randomly selected five paired LAA patients and controls on whom to conduct RNA-Seq analysis and identified 25 differentially expressed exosomal circRNAs among a pool of 462 exosomal circRNAs in total. The circRNAs were primarily from gene exons, and their lengths were mostly less than 2000 bp. In the LAA group, the expression of nine circRNAs in the exosomes was significantly increased, and sixteen circRNAs were significantly decreased (≥1.5-fold, *p* < 0.05). Subsequent GO and KEGG enrichment analysis showed that these differentially expressed exosomal circRNAs may be involved in the inflammatory response of LAA stroke, so the authors also constructed circRNA_miRNA_mRNA ceRNA (competing endogenous RNA) networks and selected two of these for validation using qRT-PCR in both LAA subjects and controls (32 versus 32). Here, they observed largely similar trends in RNA-seq as well as the significant differential expression of RNAs (circRNAs, miRNAs, and mRNAs) (*p* < 0.05). Translatable analysis showed that exosomal circRNAs contained internal ribosome entry sites, junction sequences, and open reading frames, indicating the translation potential of exosomal circRNAs. Finally, they predicted the diagnostic value of circRNAs in their validation phase by receiver operating characteristic (ROC) curve analysis and identified two circRNAs (novel_circ_001015, hsa_circ_0005585) as possible novel biomarkers for LAA stroke, and the AUC of combined circRNAs increased compared with individual ones. Taken together, these results provide information on exosomal circRNA ceRNA networks and potential encoding proteins in the progression of LAA stroke, and also showcase the possible diagnostic efficacy and biological functions of exosomal circRNA. Given that their research was aimed at mild-to-moderate LAA stroke patients (basal NIHSS score: 4.78 ± 1.00), the associations between exosomal circRNAs and different severity and subtypes of ischemic stroke warrant further study in the future.

Next, their team expanded the sample size and continued to explore the diagnostic and predictive value of exosomal circRNAs for LAA stroke. A total of 196 and 149 participants with LAA stroke and normal controls were enrolled, respectively. In the discovery experiment, they identified 26 differentially expressed circRNAs in LAA subjects compared to normal controls (|fold change| ≥ 1.5, *p* < 0.05), including 7 with significantly elevated and 19 with significantly decreased expression. In addition to the validation and replication stage, they constructed a logistic regressive model and found that circ_0043837 and circ_0001801 remained powerful and independent factors; the combined AUC of these two circRNAs was 0.825, and the diagnostic efficacy of exsomal circRNA was better than circRNA in the plasma. Furthermore, they found that exosomal circ_0043837 and circ_0001801 were potential biomarkers for predicting atherosclerotic plaque rupture; similarly, the exosomes showed superior diagnostic efficacy to plasma. The nomogram also revealed the potential predictive value of exosomal circRNAs in plaque instability. Besides this, they suggested that these two exo-circRNAs exerted a protective role in ischemic stroke, as a negative correlation between their expression and the NIHSS scores was found [21]. Considering that this was a single-center study, future multicenter studies are needed to validate these results and avoid selective bias.

All of the studies highlighted in this section showed that exosomal long noncoding RNAs and circular RNAs in the blood of ischemic stroke patients display an altered expression profile, indicating that exosomal noncoding RNAs have the potential to become both diagnostic and prognostic biomarkers in the occurrence and development of ischemic stroke.

## 4. The Involvement of Exosomal Noncoding RNAs in the Pathophysiological Process of Ischemic Stroke

In addition to monitoring the occurrence and prognosis of ischemic stroke, exosomal noncoding RNAs (lncRNAs and circRNAs) are extensively involved in the pathophysiological activities of the disease, such as atherosclerosis, inflammation, apoptosis, oxidative stress, and angiogenesis (Table 2).

### 4.1. Exosomal Long Noncoding RNAs in the Process of Ischemic Stroke

Long noncoding RNAs are a heterogeneous group of RNA transcripts greater than 200 nucleotides (nt) with a lack of long open reading frames [32]. They not only participate in several biological processes, including cell proliferation, differentiation, and chromatin remodeling but regulate gene expression at different levels, such as RNA processing, transcriptional, and posttranscriptional regulation as well. Furthermore, lncRNAs may act as miRNA ‘sponges’ or control pre-mRNA splicing [14]. In this section, we summarize the diversiform effects of exosomal lncRNAs in the progression of ischemic stroke.

#### 4.1.1. Exosomal LncRNA RMRP

One study has reported that knockdown of lncRNA RMRP (RNA component of mitochondrial RNA-processing endoribonuclease) restored the migration and proliferation of human neuroblastoma SH-SY5Y cells subjected to ischemia/reperfusion injury by suppressing autophagy and PI3K/Akt/mTOR-mediated apoptosis [33]. Zhong et al. found that changes in the noncoding RNA contents of OGD (oxygen glucose deprivation)-processed_HUVEC (human_umbilical_vein_endothelial_cell)-derived exosomes increased cell viability in vitro and decreased the infarct area in vivo by regulating PI3K/Akt/mTOR and inducing eNOS activation, and these protective effects were found to be dependent on upregulated miR-206, miR-1-3p and the distant transfer of exosomes, leading to the suppression of the expression of RMRP in neurons. The authors imitated the activity of remote ischemic post-conditioning (RIPostC) by constructing an OGD model, illustrating the orchestrated modulation of noncoding RNAs (microRNAs and lncRNAs) with mediation by exosomes [22]. Initially, the PI3K/Akt/mTOR pathway is supposed to have an anti-apoptotic effect, but Zhong et al. found instead that the inhibition of apoptosis in the neural cells corresponded with decreasing levels of p-PI3K, p-Akt, and p-mTOR. The underlying mechanisms and explicit biological behavior of RMRP and PI3K/Akt/mTOR pathway merit further investigation.

#### 4.1.2. Exosomal LncRNA MALAT1

A study by Bassit reported that lncRNA MALAT1 contained in the exosomes from human-adipose-derived stem cells (hASCs) could enhance neuronal proliferation and survival after oxidative stress induction in HT22 cells. They investigated the underlying molecular base and showed that exo-lncRNA MALAT1 promoted the alternative splicing of PKCδII (protein kinase C δII) by recruiting SRSF2 and enhancing the phosphorylation of SRSF2, leading to a neuroprotective effect. Furthermore, insulin-combined therapy increased the repairing effects of neurons by elevating the association of MALAT1 with SRSF2 [34]. Similarly, Patel et al. also found that hASC-derived exosomes and their content, lncRNA MALAT1, played an attractive role in neural regeneration after traumatic brain injury (TBI) in an in vivo rat model. Of note here is that the spleen was also found to participate in these processes of neurological function recovery, and, in doing so, provides an attractive new point of research into neural repair and regeneration. That is exosomal cargos may help to regulate spleen-related immune mediator production and release them into circulation. Another novel finding of this study was that exo-lncRNA MALAT1 could regulate not only miRNA expression but also the expression and transcription of snoRNAs in the brain; several of them were found to change in expression over twofold [35]. Another study found that lncRNA MALAT1 showed a beneficial effect on ischemic stroke by mediating anti-apoptotic and anti-inflammatory responses [36]. These studies suggest that exosomal MALAT1 might have a therapeutic potential in ischemia-induced neural damage, especially for stem-cell-derived exosomes. Finally, more extensive knowledge about underlying signal pathways and the accurate positioning of exosomal MALAT1 in the treatment of ischemic stroke is required in order to provide clinical guidance.

### 4.2. Exosomal Circular RNAs in the Process of Ischemic Stroke

Circular RNAs may be produced by precursor mRNAs’ back-splicing of exons with a covalent combination of an upstream 3′ splice site and a downstream 5′ splice site. They exhibit cell and tissue specificity with the characteristics of abundance, stability, and evolutionary conservation. They can also mediate the gene expression of mRNAs and miRNAs and regulate the translation and function of proteins, and they are also able to be translated into proteins [15]. In this section, we listed the versatile roles of exosomal circRNAs in the progression of ischemic stroke.

#### 4.2.1. Exosomal CircSHOC2

Circular RNA circSHOC2 (hsa_circ_0092670) is located at chromosome 10, whose genomic length is 13,454, with abundant expression in the cerebellum, occipital lobe, frontal cortex, and parietal lobe. A recent study from Chen’s team declared that circular RNA circSHOC2 in ischemic-preconditioned astrocyte-derived exosomes (IPAS-EXOs) could upregulate expression significantly and act as a neuroprotective agent in ischemic stroke by regulating neuronal autophagy. To probe the underlying mechanism of this, they established an OGD model in vitro and a mouse middle cerebral artery occlusion (MCAO) model in vivo and found that cellular viability was improved, as well as infarct volume and neurobehavioral deficits. The anti-apoptotic effects of exosomal circSHOC2 in neurons were reversed by this autophagic inhibitor. Continued experiments showed that exosomal circSHOC2 could be transmitted to neurons and inhibit neuronal apoptosis by promoting autophagy via sponging miR-7670-3p and subsequently increasing sirtuin 1 (SIRT1) levels, contributing to the healing of ischemia-induced brain injury. Additionally, the authors found that these pathways may be activated only in ischemia-induced environments rather than under normal conditions [23].

#### 4.2.2. Exosomal CircRNA-0006896

Yan’s group designed an experiment to determine the association between exosomal circRNAs and the formation of unstable carotid plaques, registering 22 patients with stable plaque atherosclerosis (SA) and 20 with unstable/vulnerable plaque atherosclerosis (UA), and they described the unstable plaques as lipid-rich and necrotic core plaques. In their circRNA microarray analysis and RT-qPCR validation, the concentration of exosomes and exosomal circRNA-0006896 (>2 fold) were increased differentially in the UA samples relative to the SA group (*p* < 0.001). Moreover, the upregulation of circRNA 0006896 was positively correlated with triglyceride, LDL-C (low-density lipoprotein cholesterol), and CRP (C-reactive protein) levels but negatively correlated with albumin levels, as shown in their UA-Exos group (*p* < 0.05). These cytokines are extensively involved in the progression of atherosclerosis and neuroinflammation. Accordingly, the authors built underlying circRNA-related networks by means of bioinformatic analysis and investigated functional mechanisms with an in vitro experiment using a model of HUVEC cells incubated with serum-Exos. Their results show a circRNA-0006896-miR1264-DNMT1 pathway and that the upregulation of circRNA 0006896 in serum exosomes can promote HUVEC migration and proliferation, contributing to the destabilization of carotid plaques. The authors further argued that increased expression of exo-circRNA 0006896 might decrease miR-1264 levels, leading to an elevation in DNMT1 expression and STAT3 phosphorylation, and a downregulated level of SOCS3, after the SOCS3 reduces the inhibitory effect on the JNK/STAT3 pathway and induces the formation of vulnerable plaques in the carotid artery [24]. Instability of carotid plaques is an independent risk factor for ischemic stroke, and therefore early detection and intervention in this regard can help foster more favorable outcomes. The research of Yan et al. showed that exo-circRNA-0006896 and its related network can provide novel insights into carotid-artery-related ischemic stroke; however, elaboration of the molecular bases of exo-circRNA-0006896-related pathways in ischemic stroke may yet be needed.

#### 4.2.3. Exosomal CircOGDH

CircOGDH is derived from exons 3 and 4 of the OGDH mRNA transcript. Recent research by Liu et al. demonstrated that exosomal CircOGDH might be a promising detection and therapeutic target for predicting and rescuing the ischemic penumbra tissue. In animal and human experiments, they observed significantly increased expression of exosomal CircOGDH in the ischemic mouse primary cortical neurons and plasma of AIS patients, as compared to normoxic condition and noncerebrovascular disease controls, and the level of CircOGDH in the blood was positively correlated with the size of the penumbra tissue. Mechanistically, ischemic neuron-derived exosomes transported CircOGDH from the penumbra tissue to peripheral blood, and then upregulated COL4A4 (Gallus collagen, type VI, alpha VI) expression with sequestering of miR-5112, resulting in neuronal apoptosis [25]. Their study provided a potential intervention wherein knocking down CircOGDH could enhance neuronal cell viability in the ischemic penumbra tissue, contributing to improving neurological deficits in the acute phase of ischemic stroke. However, the underlying mechanisms of the CircOGDH-miR-5112-COL4A4 pathway in neuronal apoptosis under ischemic conditions should be uncovered. The blood exosomal miR-5112 itself did not inhibit the expression of COL4A4 or the explicit biological effects of miR-5112 that also might require exploration, which also might require exploration. In addition, whether other recipient cells are involved in the cellular communication mediated by exosomes warrants investigation, as high levels of COL4A4 were seen in the ischemic astrocytes.

#### 4.2.4. Exosomal circ_0000647

Circ_0000647 generated from the gene SEC11A, is located at chromosome 15 with 380 nt mature length. Dai and colleagues discovered that the expression of exosomal circ_0000647 was elevated in oxygen-glucose deprivation and reperfusion (OGD/R)-processed SK-N-SH cells; furthermore, exo-circ_0000647 could promote OGD/R-induced damage by modulating the miR-126-5p/TRAF3 (TNF receptor associated factor 3) axis, therefore accelerating apoptosis, inflammation, oxidative stress, and inhibiting cell proliferation [26]. Their research illustrated that circ_0000647 and TRAF3 knockdown could relieve ischemia_reperfusion-induced cell injury, providing an underlying therapeutic target for ischemic stroke. Furthermore, the exosome-mediated circ_0000647/miR-126-5p/TRAF3 pathway and its pathophysiological roles in the IS progression should be confirmed in in vivo experiments, and the recipient cells of these biological effects also need more investigation.

#### 4.2.5. Exosomal circFUNDC1

CircFUNDC1 was found to display increased expression in the AIS patients, which, combined with other circRNAs, might correspond to a specificity of 91% and a sensitivity of 71.5% in AIS diagnosis [37]. CircFUNDC1, also known as hsa_circ_0007290, is derived from the FUN14 domain containing 1 (FUNDC1) mRNA. Bai’s study discerned that increased circFUNDC1 levels in the serum-derived exosomes of AIS patients and presented exosomal circFUNDC1 could be a potential diagnostic index for IS, with a value of 0.8822 for the area under the ROC curve. They also found that circFUNDC1 knockdown alleviated OGD-induced BMEC injury by suppressing PTEN (phosphatase and tensin homolog) by removing the sponge effects on miR-375 [27]. The biological roles of exosomes in circFUNDC1/miR-375/PTEN are ill-defined so far; more in vivo experiments are needed to clarify the role of the exosomal circFUNDC1 pathway in IS progression.

#### 4.2.6. Exosomal CircAkap7

CircAkap7 is located at chromosome 10 and derived from exon 2 of gene Akap7 with a length of 579 base pairs in its spliced mature sequence. Xu’s group identified the therapeutic effects of exosomes derived from circAkap7-modified ADSCs (exo-circAkap7) in the mouse tMCAO model in vivo and OGD/R model in vitro, while exosomes alone did not display similar effects. Their results showed that circAkap7 could be delivered by exosomes to the cortex and ameliorate cerebral injury by reducing the infarct volume and improving sensorimotor dysfunction. Similar beneficial effects were also shown in an in vitro experiment involving co-culture with exo-circAkap7 and OGD-processed primary astrocytes. Further investigation indicated the exo-circAkap7 inhibited the levels of miR-155-5p, and then promoted ATG12-mediated autophagy, as well as suppressing NRF2-mediated oxidative stress and inflammation [28]. These findings provide a novel target for the early phase of ischemic stroke therapy (up to 7 days after stroke) based on exosomal circAkap7, which could be meaningful for the functional recovery and acute management of IS.

#### 4.2.7. Exosomal Circ-Rps5

Another study by Yang et al. [29] found that exosomal circ-Rps5 derived from hypoxic pre-treated ADSCs (adipose-derived stem cells) could enhance acute ischemic stroke-induced cognitive function by reducing neuronal injury. Mechanistically, exosomes mediated the delivery of circ-Rps5, later sponged downstream miR-124-3p, and increased the expression of SIRT7, which ultimately promoted microglia to the M2-phenotype in the hippocampus and exerted anti-inflammatory and anti-apoptotic effects.

In this section, we have enumerated and summarized the widespread involvement of exosomal lncRNAs and circRNAs in the occurrence and development of ischemic stroke, respectively. Noticeably, several types of noncoding RNAs participate in the pathophysiological processes of ischemic stroke in a reciprocal regulation form. Given that the comprehensive involvement of exosomal noncoding RNAs in ischemic stroke has yet to be fully addressed, especially for lncRNAs and circRNAs, further investigation of their crosstalk within the disease seems to be a logical next step in the near future. We depict explicit exosome-mediated noncoding RNAs crosstalk in the process of ischemic stroke in Figure 1.

## 5. Recent Advances in Exosome Applications in the Treatment of Ischemic Stroke

Currently, there are still some deficiencies in traditional treatment strategies for ischemic stroke, consisting primarily of hemorrhagic transformation, reperfusion injury, and a limited time window for treatment, and the existence of the blood-brain barrier (BBB) makes targeted therapy quite difficult as well. During ischemic stroke, damage to the BBB forces the brain to confront vast inflammatory factors, pathogens, and toxins directly, which can cause cerebral parenchyma injury. The stabilization and reconstruction of the integrity of the BBB is a viable treatment option for ischemic stroke, and the findings from an experiment by Zhang suggest that neural-progenitor-cell-derived extracellular vesicles can enhance poststroke BBB integrity via the inhibition of the NF-κB signaling pathway in both endothelial cells and astrocytes, contributing to the mitigation of the upregulation of ABCB1 and MMP-9. The underlying mechanism causing these EVs to suppress NF-κB activation needs more investigation, however. During the past two decades, nanoscale brain-targeting therapeutic strategies have come into view and may offer the advantages of being modifiable and having monitored delivery vectors for targeted transportation [38,39].

Extracellular vesicles such as exosomes, with their natural characteristics of circulation stability and ability to pass through the BBB, along with the capacity for encapsulating and protecting endogenous bioactive molecules, have several advantages over synthetic nanocarriers for drug delivery. First, exosomes can be engineered to fit specific requirements for delivery within their lipophilic bilayers or aqueous cores. Furthermore, they are internalized by recipient cells in various ways, such as endocytosis, direct fusion with cell membranes, and ligand_receptor interaction [40]. Another essential feature of exosomes is that they can be preserved for a long time at −80 °C and can survive other extremes of handling. Finally, exosomes can pass through the BBB and protect their cargo from degradation [41]. Therefore, exosomes may be an ideal vehicle for transmitting contents to recipient cells in the CNS (central nervous system).

As for ischemic stroke, exosome-related treatments are instrumental in many stages of the disease [42]. Above all, exosomes may be able to transmit thrombolytics to specific sites effectively by means of physical targeting, ligand-mediated active targeting, and biomimetic targeting. Furthermore, since ischemia-induced cerebral damage is a complex pathophysiological process, exosome-related interventions could be implemented at different stages of stroke, such as for the correction of hypoxemia and out-of-balance ions, the reduction of oxidative stress and apoptosis, the regulation of inflammatory and immune responses and the promotion of tissue repair. In addition, exosomes can offer a neuroprotective effect in post stroke patients via endothelial_neuronal matrix coupling targeted towards the BBB. The mediation of ischemia-induced downstream cascades may be able to achieve more favorable outcomes for inflammation, apoptosis, and neural repair since upstream events may take place too quickly to be reversed by treatment after the fact [43]. Furthermore, stem-cell-derived exosomes have demonstrated the capability to aid in functional recovery after injury induced by ischemia [44]. However, another beneficial feature of the exosome-mediated treatment is exosomes’ high selectivity for transferring cargo to appointed sites (for example, the ischemic penumbra), which may elevate the effective utilization of bioactive substances, decrease minimum effective dosages, and also decrease toxicity, though nontargeted drugs may still be preferred for accumulation in the ischemic core where the more serious deconstruction of the BBB has occurred. Finally, recent biomimetic technology has succeeded in establishing exosomes based on living cells or cell membranes, guaranteeing targeted transportation and reducing immunogenicity at the same time [7,43,45].

More specifically, one recent study successfully constructed cyclo (Arg-Gly-Asp-D-Tyr-Lys) peptide-conjugated exosomes (cRGD-Exo) as an effective targeting delivery vehicle for ischemic stroke [46], and another delivery system was established by Zhang. Zhang’s novel cl PGP-PEG-DGL/CAT-Aco nanoparticles (NPs) were able to carry macromolecular drugs to inflammatory sites and reduce the infarct volume of MCAO mice by repressing ROS-mediated apoptosis [47]. In addition, Kim et al. found that iron_oxide_nanoparticle_(IONP)_harboring_MSC-derived nanovesicles that possessed magnetic properties could increase the targeting ability to ischemic lesions and promote the therapeutic outcome through anti-apoptotic, angiogenic and anti-inflammatory effects [48] (Table 3).

In summary, research has showcased the considerable potential of exosome-related therapeutic measures during different periods of ischemic stroke. Exosomes can cross the BBB to target recipient cells in the brain more effectively than other treatments through the peripheral circulatory system. Future research that probes into methods of delivery to specific and targeted cells in diverse areas such as the ischemic penumbra is likely to produce many new breakthroughs in the treatment of ischemic stroke.

## 6. Therapeutic Prospects of Exosomal Noncoding RNAs in the Treatment of Ischemic Stroke

In recent years, the application of gene therapy has become more widespread due to its ability to implement precise and individualized treatment for hereditary diseases, cancer, and neurodegenerative diseases. Gene therapy is the therapeutic delivery of genetic material to recipient cells in order to correct the abnormal expression of genes or to produce beneficial proteins that promote protection or recovery. With the recent FDA approval of patisiran, the first RNAi drug delivered by lipid nanoparticles, RNA-based gene therapy has also made great progress in clinical settings. The most concerning issue for gene therapy is its safety; therapists must seek efficient and low-immunogenicity carriers to targeted recipient cells in order to minimize dosages. Exosomes, as live cell secretory nanovesicles, can effectively transport nucleic acids such as noncoding RNAs to targeted cells and thereby help them to exert specific biological effects [53,54].

Recent advances in the modification techniques of exosomes have created a favorable condition for tailored delivery in ischemic brains. Tian’s team designed engineered EVs derived from ReNcell VM cells (RGD-EVReN) with the decoration of Arg-Gly-Asp (RGD)-4C peptide and phosphatidylserine-binding domains of lactadherin (C1C2) that were observed to manifest high tropism to ischemic brain lesions and anti-inflammatory efficacy by suppressing TNF-α, IL-1β and IL-6 levels in a mouse MCAO model. In their findings regarding miRNA sequencing and following validation, they found that the inhibition of the microRNAs-mediated mitogen-activated protein kinase (MAPK) signal pathway could be a contributor to this process. They also pointed out that the modification and systemic administration of the EVs did not influence their endogenous bioactivity, and they observed no obvious side effects after injection [49]. In addition, due to their ability to be rapidly produced, these functionalized EVs showed promise for clinical applications, including gene therapy. Another study found that interferon-gamma-preconditioning-human-neural-stem-cell-derived exosomes (IFN-γ-hNSC-Exo) exerted better neuroprotective and neurofunctional therapeutic effects on ischemic stroke than unprocessed hNSCs-Exo in a rat model and that exosomal miRNAs were important mediators of cell survival, especially hsa-miR-206, hsa-miR-133a-3p and hsa-miR-3656 [50]. Furthermore, Zhang et al. reported that MSC-derived exosomal miR-210 with specific decoration could be targeted to ischemic lesions effectively and could improve microvascular angiogenesis in a murine MCAO model. The exosomes were conjugated on cyclo(Arg-Gly-Asp-D-Tyr-Lys) peptide (RGD-exo) and then incubated with cholesterol-modified miR-210 to construct RGD-exo:miR-210, which was administered via the tail vein once every other day for 14 days, resulting in increased VEGF expression and vascular density remaining high for two weeks [51]. Furthermore, these exosome-based therapeutic measures for ischemic stroke could be modified to amplify the mediating effects of RNAs, which illustrates that exosomal RNAs hold promise as preclinical interventions and feasible clinical treatments.

Besides the above, Alvarez-Erviti et al. showed that siRNA (small interfering RNAs) could be loaded into exosomes by an electroporation protocol and that these exosomes fulfilled the specific gene knockdown of neurons, microglia, and oligodendrocytes in the brain of mice after being intravenously injected [55]. The current belief is that exosomes are ideal RNA targeting delivery vehicles with high biocompatibility and modificability, and low immunogenicity and clearance rate by macrophages as well [56]. Making use of these benefits, Kim et al. designed exosomal siRNA-mediated therapeutic measures for ischemic stroke in a rat MCAO model. They constructed rabies_virus_glycoprotein_(RVG) peptide-decorated exosomes to transport HMGB1- (high-mobility group-box-1) siRNA to ischemic rat brains by intravenous administration through the tail vein, and they established a brain ischemic rat model after 18 h of injection. This delivery method induced the downregulation of HMGB1 and decreased levels of tumor necrosis factor-α (TNF-α) and apoptosis, leading to reduced infarct volume [52]. Moreover, Zhang’s team established a self-assembly strategy of siRNAs in vivo with safety and effectivity and delivered mHTT-silencing siRNA to the cortex and striatum based on a rabies virus glycoprotein-tagged exosome-circulating system, ameliorating behavioral deficits in Huntington’s disease [57]. Meanwhile, the long-term influence of repeated administration on the liver function and exploration of the half-lives, distribution, and metabolism of exosomal siRNAs are conducive to the assessment of the safety and optimal dosage of exosomal RNAs delivery, which is the foundation of clinical translation [58]. Their findings demonstrated directly that exosomes could be siRNA carriers across the BBB and could give rise to fast and transient responses in their recipient regions with high efficacy and low toxicity, which indicates a new research direction in the time-dependent treatment of ischemic stroke.

Yang’s team established EV-mediated circular RNA transfer and observed their effects after ischemia using both rodent and nonhuman primate models. Their targeted carriers to injured regions were engineered by fused protein Lamp2b (lysosome-associated membrane glycoprotein 2b)-RVG and incorporated into EVs with circSCMH1 overexpression (RVG-circSCMH1-EVs), which is a type of circRNA derived from exons 6, 7, and 8 of the SCMH1 (Scm Polycomb Group Protein Homolog 1) gene. After intravenous administration of RVG-circSCMH1-EVs, they discovered improved neurological functions via elevated neuroplasticity and declining glial activation and peripheral immune cell infiltration. Moreover, the combination of circSCMH1 and MeCP2 (methyl-CpG binding protein 2) relieved the suppression of downstream gene transcription (Mobp, Igfbp3, Fxyd1, and Prodh) [30]. More practically, this potential therapeutic strategy could be implemented at 24 h after stroke onset, implying a broader time window for the therapy of ischemic stroke.

Another innovative therapeutic strategy was designed by Kim et al., who produced RBP-Exo using exosomes connected with RAGE- (receptor for advanced glycation end-products) binding_peptide as a hypoxia-specific carrier to deliver cholesterol-modified AMO (anti-microRNA oligonucleotide)-181a to the ischemic brain. Here, the intranasal administration of RBP-Exo/AMO181a-chol integrated the neuroprotective effects of RBP (also named recombinant RAGE-antagonist peptide) and anti-miR-181a, and decreased the infarct volume by inhibiting inflammation and apoptosis post stroke [31]. Their study provided a novel carrier for RNA_transfer as well as an intranasal method for the entrance of exosomes to the ischemic brain, contributing to developments in both genetic engineering and therapy for ischemic stroke.

However, there are also other existing potential ventures in the clinical application of exosomes besides their therapeutic advantages. Exosomes are reported to induce molecular alteration in cells under certain situations; furthermore, supraphysiological levels of exosomes might be associated with the modulation of cancer progression and induction of neoplasia in mice [6]. There is an ongoing phase I clinical trial on the therapeutic effects of siRNA_loaded_MSC_derived exosomes in cancer (NCT03608631), and for ischemic stroke, a randomized, single-blind, placebo-controlled phase I/II clinical trial is also in progress at the time of writing (NCT03384433). The aim of the latter trial is to assess the safety and efficacy of the exosomes in improving the disability among AIS patients who receive total protein of allogenic-MSC-generated exosomes transfected by miR-124 via stereotaxic/intraparenchymal methods, one month after the attack. However, the exploration of the optimal effective dosage and administration time, as well as method, in stroke therapy is also needed in future research.

## 7. Future Work and Perspectives

We have described the latest progress in the use of exosomes combined with noncoding RNAs to address the occurrence, progression, and recovery of ischemic stroke. Furthermore, we also portray the altered expression profiles of exosomal noncoding RNAs subjected to ischemia in different biological models, showcasing their diagnostic value in general. In the last section of this review, we discuss the potential therapeutic efficiency of exosomes and noncoding RNAs in the current research and attractive directions for research in the future. While the investigation of exosomes and their contained noncoding RNAs is still in its infancy, we believe that it will be of great future interest to examine the roles of a variety of noncoding RNAs enriched in different cell-derived exosomes in acting on intercellular crosstalk. Exosomes could build a bridge of communication between peripheral circulation and the CNS, and noncoding RNAs are versatile participants in the pathophysiological process of ischemic stroke. That is, continued exploration of exosomal noncoding RNAs in the treatment of ischemic stroke carries with it important biological and clinical significance.

In conclusion, with advances in clinical and experimental technologies, we expect exosomal noncoding RNAs to become the ideal circulating biomarkers for use in the diagnosis and monitoring of ischemic stroke. Due to their numerous roles in the occurrence and progression of the disease, exosomal noncoding RNAs offer interesting leads for the development of novel therapeutic targets. However, existing exosome-based approaches still require additional evaluation of their side effects for clinical translation. Overall, we are optimistic about the implementation of novel diagnostic and therapeutic measures based on exosomal noncoding RNAs in clinical applications.

## Figures and Tables

**Figure 1 brainsci-12-01000-f001:**
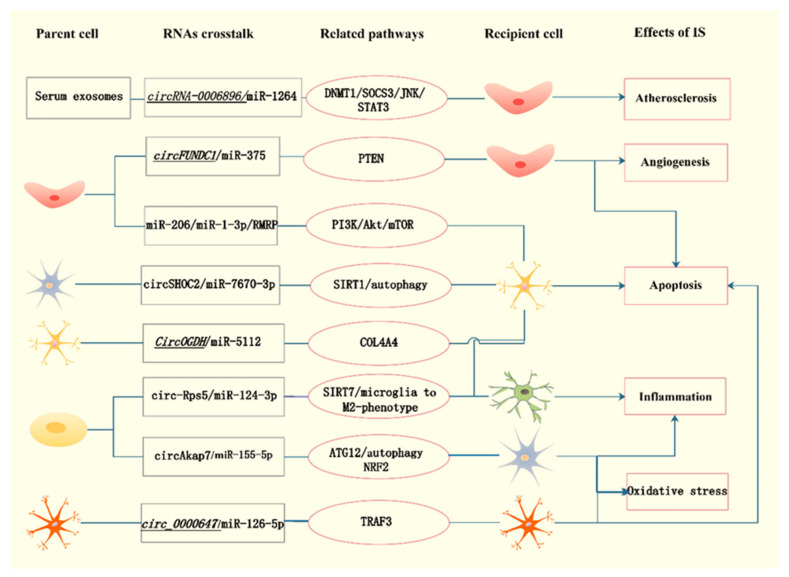
Exosomal noncoding RNAs crosstalk in the pathophysiological process of ischemic stroke. Figure annotation: 
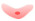
 endothelial cell 
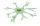
 migroglia 
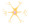
 neuron 
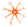
 SK-N-SH cells 
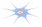
 astrocyte 
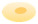
 adipose-derived stem cell. (The italic represents detrimental effect of stroke).

**Table 1 brainsci-12-01000-t001:** Differential expression profiling of exosomal noncoding RNAs in ischemic stroke.

Models and Tissues	Detection Method	Changes of Expression Profiles	Focal ncRNAs	Reference
Exosomes from peripheral serum in stroke patients	RNA-Seq and qRT-PCR	1096 lncRNAs covering 307 showed elevated expression while 789 showed decline	lnc-CRKL-2 and lnc-NTRK3-4 ↑ RPS6KA2-AS1 and lnc-CALM1-7 ↓	Xu [16]
Exosomes from peripheral plasma in stroke patients	HTS and qRT-PCR	1020 lncRNAs were differentially expressed, 226 lncRNAs increased and 794 lncRNAs decreased levels	Most: novel lnc_000288 ↑; novel lnc_000285 ↓ lnc_002015/hsa-mir342/PTPRC/AAMP; lnc001350/hsa-mir-3127/ST14	Zhang [17]
Exosomes from peripheral plasma in stroke patients	RNA-Seq and qRT-PCR	319 lncRNAs including 97 increased expression and 222 showed reduction	lnc_000048, lnc_001350 and lnc_016442 ↑ lnc_002015 ↓	Zhang [18]
Exosomes from peripheral plasma in stroke patients	HTS	3540 circRNAs, 1177 increased expression and 2363 decrease expression	/	Xu [19]
Exosomes from peripheral plasma in stroke patients	RNA-Seq and qRT-PCR	25 circRNAs, 9 circRNAs were significantly upregulated and 16 circRNAs were significantly downregulated	hsa_circ_0000698/hsa_circ_0002775/hsa_circ_0005585/hsa_circ_0043837 ↓; hsa-miR-16 ↑; VWF ↓; novel_circ_0010155 ↓; hsa-miR-939 ↑; septin 9 and MYLK2 ↓	Xiao [20]
Exosomes from peripheral plasma in stroke patients	RNA-Seq and qRT-PCR	26 circRNAs, 7 circRNAs were significantly upregulated and 19 circRNAs were significantly downregulated	circ_0043837 circ_0001801	Xiao [21]

↑ upregulation; ↓ downregulation.

**Table 2 brainsci-12-01000-t002:** The source, expression, and mechanisms of exosomal noncoding RNAs in ischemic stroke.

Source	RNA	Expression	Target/Mechanisms	Function/Effects of Dysregulation	References
Exosomes from HUVECs subjected to OGD	lncRNA RMRP	/	miR-206, miR-1-3p and RMRP/PI3K/Akt/mTOR pathway	Increased cell viability and decreased infarct volume by regulating apoptosis and inducing eNOS activation	Zhong [22]
Exosomes from ischemic-preconditioned astrocytes	circSHOC2	Upregulation	circSHOC2/miR-7670-3p/SIRT1	Decreased infarct volume and neurobehavioral deficits by reducing neuronal apoptosis via regulating autophagy	Chen [23]
Serum exosomes from patients with carotid plaque	circRNA-0006896	Increased levels in UA group	circRNA-0006896-miR1264-DNMT1/SOCS3/JNK/STAT3 axis	Promoted endothelial cell migration and proliferation, induced plaque destabilization	Yan [24]
Exosomes from ischemic mouse primary cortical neurons and plasma of AIS patients	CircOGDH	Upregulation	CircOGDH/miR-5112/COL4A4	Knockdown relieved neuronal injury by inhibiting apoptosis The expression was positively related to the size of penumbra in AIS patients	Liu [25]
Exosomes from OGD/R-induced SK-N-SH cells	circ_0000647	Upregulation	circ_0000647/miR-126-5p/TRAF3	Accelerate apoptosis, inflammation, oxidative stress and inhibit cell proliferation	Dai [26]
Serum exosomes from AIS patients	circFUNDC1	Upregulation	circFUNDC1/miR-375/PTEN	Knockdown promoted cell survival and angiogenesis	Bai [27]
Exosomes from ADSCs	circAkap7	/	circAkap7/miR-155-5p/ATG12, NRF2	Protected against cerebral ischemic injury by promoting autophagy and inhibiting oxidative stress	Xu [28]
Exosomes from ADSCs	cir_Rps5	Upregulation	cir_Rps5/miR-124-3p/SIRT7 axis	Improved ischemic induced cognitive function via decreasing neuronal damage in the hippocampus	Yang [29]
RVG-circSCMH1-EVs	circular RNA SCMH1	/	circSCMH1/MeCP2/Mobp, Igfbp3, Fxyd1, and Prodh	Enhanced functional recovery including promotion of brain plasticity, reduced glial activation, and peripheral immune cell infiltration	Yang [30]
RBP-Exo/AMO181a-chol	miR-181a	/	Bcl-2	Decreased infarct volume by inhibiting inflammation and apoptosis	Kim [31]

**Table 3 brainsci-12-01000-t003:** Recent advances in Exosome Applications in the Treatment of Ischemic Stroke.

Origin	Cargo	Function	Target	Implication	Ref.
Neural progenitor cell	/	Inhibited NF-κB signaling pathway	ECs, astrocytes	Enhanced BBB integrity poststroke	Zhang [38]
Mesenchymal stromal cell	Curcumin	Biochemically engineered cRGD-Exo	Microglia, neurons, astrocytes.	Inhibited inflammation and cellular apoptosis	Tian [46]
Aqueous suspension	/	Constructed cl PGP-PEG-DGL/CAT-Aco NPs ROS-mediated apoptosis	Neuronal cells	Reduce infarct volume through suppressing inflammation	Zhang [47]
Mesenchymal stem cell	Therapeutic growth factors	Constructed MSC-IONP-derived magnetic nanovesicles	ECs, neuronal cells, macrophages	Decreased infarction volume and improved motor function via angiogenesis, anti-inflammation and anti-apoptosis	Kim [48]
Neural progenitor cell	MiRNAs	Constructed RGD-EV MAPK signal pathway	Microglia	Improved targeting ability to the ischemic lesion and suppressed poststroke inflammation	Tian [49]
Neural stem cell	MiRNAs	Constructed IFN-γ-hNSC-Exo	Neurons, microglia	Reduced infarct volume and facilitated the neurological functional recovery by increasing cell proliferation and survival, decreasing cell apoptosis and inflammation	Zhang [50]
Mesenchymal stromal cell	MiR-210	Constructed RGD-Exo	ECs	Improved microvascular angiogenesis by upregulating VEGF expression	Zhang [51]
HEK293T cells	HMGB1- siRNA	Constructed RVG-Exo	Neuro2A cells	Reduced infarct volume by suppressing TNF-α expression and cell apoptosis	Kim [52]

## Data Availability

Not applicable.

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
