# Peer review of "A Novel Perspective on Ischemic Stroke: A Review of Exosome and Noncoding RNA Studies"

_brainsci, 2022, doi:10.3390/brainsci12081000_

Round 1

Reviewer 1 Report

The manuscript titled “A Novel Perspective on Ischemic Stroke: A Review Exosome and Noncoding RNA Studies” the authors describe the latest progress in the use of exosomes combined with noncoding RNAs in terms of occurrence, progression, and recovery for ischemic stroke.

This manuscript is not acceptable for publication in the present form. The manuscript lacks comprehensive view of the field of research, including a more in-depth analysis of the work in the field around the globe. The authors should include a section in which they describe the methodology they used for literature survey and the selection criteria for inclusion of studies related to ischemic stroke.  In two instances, the authors cite retracted papers (ref no 22 &25), one even having its own section. This is a serious laps in judgement. Another concerning issue is the lack of references, especially when describing properties of exosomes. For example, the entire second paragraph that introduces exosomes contains only one reference! The same issue continues throughout the manuscript. The authors need to carefully add appropriate references to the text.  

The writing is often imprecise and ambiguous.

The manuscript titled “A Novel Perspective on Ischemic Stroke: A Review Exosome and Noncoding RNA Studies” the authors describe the latest progress in the use of exosomes combined with noncoding RNAs in terms of occurrence, progression, and recovery for ischemic stroke.

This manuscript is not acceptable for publication in the present form. The manuscript lacks comprehensive view of the field of research, including a more in-depth analysis of the work in the field around the globe. The authors should include a section in which they describe the methodology they used for literature survey and the selection criteria for inclusion of studies related to ischemic stroke.  In two instances, the authors cite retracted papers (ref no 22 &25), one even having its own section. This is a serious laps in judgement. Another concerning issue is the lack of references, especially when describing properties of exosomes. For example, the entire second paragraph that introduces exosomes contains only one reference! The same issue continues throughout the manuscript. The authors need to carefully add appropriate references to the text.  

The writing is often imprecise and ambiguous.

Title “A Novel Perspective on Ischemic Stroke: A Review Exosome and Noncoding RNA Studies” please add “OF exosome and  Noncoding RNA Studies”

 Introduction

line 37-40 Reconsider the definition of exosome and rephrase the sentence to make it more precise - it implies that exosomes are divided into two groups. Please consult ISEV 2018 recommendations for extracellular vesicle nomenclature.

Line 42 Rephrase the “double invagination” formulation. This is not accurate enough to describe to biogenesis process.

Line 46 Debatable statement, it is hardly true that the research is in its infancy. Furthermore, the authors state in the next section that there’s “an array of evidence” of the utility of exosomes as biomarkers in different disease.

Section 2.

Line 126-128 Rephrase, differentially expressed genes map to a chromosome

Line 128. Please define the hub genes

Line 171 rephrase “elevated expressions” and “declined levels”

Line 225. Rephrase

Line 227-230 This statement by the authors is not corroborated by the studies presented – they only illustrate biomarker potential and no biological processes (including atherosclerosis, neuro-inflammation, immunity, apoptosis, and neural repair) are mentioned.

Section 3.

Line 242 Please rephrase “Long noncoding RNAs are defined as linear noncoding RNA”

Line 250 Change “on” to “one”

Line 252/262 Do you mean “neurons” instead of “nerve cells” or “neural cells”?

Line 262 Please replace “lucid” with a more appropriate word

Section 3.2.2. This article has been retracted and this entire section needs to be removed.

Section 4.

 Please replace “Exosome-Mediated Applications” with “exosome applications”.

Line 469. Please rephrase. Do you mean endogenous or therapeutic exosomes? Therapeutic exosomes are for instance cleared very rapidly after i.v. injection.

Line 480. More information needs to be given about targeted exosome delivery to make this statement. The authors MUST include appropriate references when describing a term, effect or making any claims new to the text.

Line 488. Reference

Line 525. “concerning” not “concerned”

Line 558. The cited study did not investigate safety of the applied treatment, only off-target uptake

Line 561. Not sure what this means, certainly requires a reference

572-573 How are these properties conductive to clinical translation, please specify

602. “including the trigger of neurodegenerative disorder,” this means the exosomes initiate the disease. Is this what you intended to say?

603. Reconsider the use of tense, these trials are ongoing

Line 613. Change “prospection” to another appropriate word. 

Author Response

Dear Editors and Reviewers:

Thank you for your comments concerning our manuscript. Those comments are all valuable and very helpful for revising and improving our paper. We have studied reviewer’s comments carefully and have made revision which marked in red in the paper. 

We are very sorry for our imprecise writing. The text had edited by a native speaker and have been corrected typo's and grammatical errors. We appreciate for your warm work earnestly, and hope that the correction will meet with approval. If you don’t agree with us and we would like to revise our manuscript again.

Reviewer 2 Report

In the manuscript A Novel Perspective on Ischemic Stroke: A Review Exosome and Noncoding RNA Studies” the authors have tried to review role of exosomal RNAs and its biological/therapeutic effects in the progression of ischemic stroke. Though the content of the review article looks great, I have some concerns that need to be addressed.

  1. Overall, the review is overwhelmed with a vast description/explanation of single report. In many places, a single long paragraph or even sections are dedicated to highlight or explain the findings from a particular study. This kind of narrative is not necessary and need to be simplified with clear and concise reports. Authors should revisit this and modify the style of writing throughout the manuscript.
  2. Title of the review is not clear and needs to be corrected.
  3. Abstract looks like a small introduction. This need to be summarized with a concise description of the essence of this particular review article at least at the end.
  4. In many places in the review, authors should expand the short form wherever it is mentioned for a better understanding in a reader’s perspective (Eg-NIHSS, GO etc). Also, please correct the typo throughout the manuscript (Eg-In line 250-On study….).
  5. It would be great if authors could remove the GO analysis and ROC curve column from Table 1. Also, place the Reference towards the last column. These changes would be helpful not to overcrowd the table. Also, this will convey the essential details with respect to the title of the table.
  6. In Table 2, inclusion of lnc RNA MALAT1 is missing. This should be included.
  7. It would be great if authors could generate a table that highlight the recent advances in Exosome-Mediated Applications in the Treatment of Ischemic Stroke.

Author Response

Dear Editors and Reviewers:

Thank you for your comments concerning our manuscript. Those comments are all valuable and very helpful for revising and improving our paper. We have studied reviewer’s comments carefully and have made revision which marked in red in the paper. 

Round 2

Reviewer 2 Report

Thank you for revising the manuscript. It looks great now.

Author Response

We appreciate for your warm work earnestly, and hope that the correction will meet with approval. If you don’t agree with us and we would like to revise our manuscript again.

This manuscript is a resubmission of an earlier submission. The following is a list of the peer review reports and author responses from that submission.